# Overexpression of *AtBBD1*, Arabidopsis Bifunctional Nuclease, Confers Drought Tolerance by Enhancing the Expression of Regulatory Genes in ABA-Mediated Drought Stress Signaling

**DOI:** 10.3390/ijms22062936

**Published:** 2021-03-13

**Authors:** A. K. M. Mahmudul Huque, Wonmi So, Minsoo Noh, Min Kyoung You, Jeong Sheop Shin

**Affiliations:** 1Division of Life Sciences, Korea University, Seoul 02841, Korea; mahmudornob@korea.ac.kr (A.K.M.M.H.); thso1124@gmail.com (W.S.); redmoon89@korea.ac.kr (M.N.); 2Graduate School of Biotechnology, Kyung Hee University, Yongin 446-701, Korea

**Keywords:** *Arabidopsis thaliana*, *AtBBD1*, DUF151 domain, ABA response, abiotic stress, drought tolerance, stomatal movement

## Abstract

Drought is the most serious abiotic stress, which significantly reduces crop productivity. The phytohormone ABA plays a pivotal role in regulating stomatal closing upon drought stress. Here, we characterized the physiological function of AtBBD1, which has bifunctional nuclease activity, on drought stress. We found that AtBBD1 localized to the nucleus and cytoplasm, and was expressed strongly in trichomes and stomatal guard cells of leaves, based on promoter:GUS constructs. Expression analyses revealed that *AtBBD1* and *AtBBD2* are induced early and strongly by ABA and drought, and that *AtBBD1* is also strongly responsive to JA. We then compared phenotypes of two *AtBBD1*-overexpression lines (*AtBBD1*-OX), single knockout *atbbd1*, and double knockout *atbbd1/atbbd2* plants under drought conditions. We did not observe any phenotypic difference among them under normal growth conditions, while OX lines had greatly enhanced drought tolerance, lower transpirational water loss, and higher proline content than the WT and KOs. Moreover, by measuring seed germination rate and the stomatal aperture after ABA treatment, we found that *AtBBD1*-OX and *atbbd1* plants showed significantly higher and lower ABA-sensitivity, respectively, than the WT. RNA sequencing analysis of *AtBBD1*-OX and *atbbd1* plants under PEG-induced drought stress showed that overexpression of *AtBBD1* enhances the expression of key regulatory genes in the ABA-mediated drought signaling cascade, particularly by inducing genes related to ABA biosynthesis, downstream transcription factors, and other regulatory proteins, conferring *AtBBD1*-OXs with drought tolerance. Taken together, we suggest that AtBBD1 functions as a novel positive regulator of drought responses by enhancing the expression of ABA- and drought stress-responsive genes as well as by increasing proline content.

## 1. Introduction

Plants are constantly challenged by various abiotic stresses, including UV irradiation, high temperature, cold, high salinity, and drought, and they exhibit rapid changes at both physiological and molecular levels to adapt to changing environmental conditions. Of these abiotic stresses, drought is the most serious abiotic stress. Drought causes severe damage to plants and significantly reduces crop productivity. While the underlying adaptive mechanisms in response to drought stress have been extensively investigated [1], the complex and synchronized functional changes in planta driven by drought stress are not yet properly understood.

Several survival mechanisms to confront drought stress have evolved in plants, such as accumulation of osmoprotectants like proline, maximizing water uptake, as well as stomatal closing to minimize transpirational water loss. The phytohormone ABA is a key regulator for drought stress responses in plants. Drought conditions induce the expression of ABA biosynthetic genes, leading to accumulation of ABA [2]. ABA triggers a signaling cascade in the stomatal guard cells in response to the drought stress, resulting in stomatal closure and reduced transpiration water loss, thereby reducing drought stress [3,4]. It is well known that many drought-responsive regulatory proteins, such as transcription factors (TFs) (AREB, AP2/ERF, NAC, bZIP, MYC, and MYB), protein kinases (MAPK, CDPK, receptor protein kinases, ribosomal protein kinases, and transcription regulation protein kinases) and protein phosphatases (PP2Cs), are crucial for plants’ defense to the drought stress.

Jasmonic acid (JA) is also associated with the alleviation of drought stress in plants. A subtle network of ABA-JA hormonal crosstalk related to physiological responses to drought has emerged in the last decade. For example, *Arabidopsis thaliana* MYC2/JIN1 (Jasmonate insensitive 1), which is the core regulator in JA signaling [5], was initially characterized as a positive regulator of ABA signaling, modulating drought tolerance in Arabidopsis through regulating *RD22* (*Response to Desiccation 22*) [6], *ANAC019* (*NAC domain-containing protein 19*), and *ANAC055* (*NAC domain-containing protein 55*) [7]. In rice, OsJAZ1 functions as a negative regulator of drought stress response by negatively regulating ABA and JA responses [8]. Similarly, MYB44 (MYB domain proteins 44), which confers abiotic stress tolerance by repressing the expression of negative regulators of ABA signaling, such as *ABI1*, *ABI2*, *AtPP2CA*, *HAB1*, and *HAB2* [9], was also reported as a repressor of JA-mediated defense response through regulation of *WRKY70* [10]. Furthermore, CML37 (calmodulin like 37), a Ca^2+^ sensor that is involved in jasmonate-mediated defense response against herbivores [11], was also identified as a positive regulator of ABA during drought stress response [12].

Nucleases in higher plants play pivotal roles in many crucial physiological processes, including senescence [13], programmed cell death (PCD) [14], development [15], and biotic and abiotic stress responses [16,17]. While a number of studies reported the enhancement of nuclease activity in response to abiotic stress, such as salt stress [18,19], direct evidence regarding the involvement of nucleases in modulating abiotic stress has been rarely reported. Recently, Sui et al. (2019) reported that AtCAN2 (calcium dependent nuclease 2), a Ca^2+^-dependent DNase-RNase, plays a negative role in responses to salt stress in Arabidopsis [16]. Another study reported that a direct association of drought stress-induced PCD, with nucleic acid degradation by the enhanced hydrolase activity, indicates the possible involvement of nucleases in drought stress responses [20]. However, no other study has been reported that demonstrates a role for nucleases in regulation of drought stress response.

We recently reported that the DUF151 domain-containing Arabidopsis BBD (Bifunctional nucleases in Basal Defense response) proteins, AtBBD1 and AtBBD2, exhibit non-substrate-specific DNase and RNase activity [21], similar to OmBBD in *Oryza minuta*, possibly involving an ABA-mediated synergistic interaction between ABA and JA against biotic stresses [17]. In this study, we characterized the physiological function of AtBBD1. We checked the expression patterns of *AtBBD1* in six different tissues, and their expression after treatment with phytohormones and abiotic stresses. Then, we compared the phenotypes of *AtBBD1*-OXs and *atbbd1* with WT under both water-limited treatments in soil and polyethylene glycol (PEG)-induced drought stress conditions. Subsequently, we confirmed that AtBBD1 is as a positive regulator in ABA-mediated drought response.

## 2. Results and Discussion

### 2.1. AtBBD1 Localizes to the Nucleus and Cytoplasm, and Is Strongly Expressed in Trichomes and Stomatal Guard Cells of Leaves

To determine the subcellular localization of AtBBD1 and AtBBD2, *Pro_35S_:BBD1-sGFP* and *Pro_35S_:BBD2-sGFP*, constructs were introduced into Arabidopsis protoplasts. We observed that both BBD1-sGFP and BBD2-sGFP signals overlapped with NLS-RFP in the nucleus and also localized to the cytosol (Figure 1A). 

Then, we investigated the tissue-specific expression patterns of *AtBBD1* and *AtBBD2* genes in five different tissues of four-week-old plants; rosette leaf, cauline leaf, stem, flower, and root, using qRT-PCR. We found that the expression of *AtBBD1* was much higher than that of *AtBBD2* in all tissues except in root, and that *AtBBD1* was expressed more than three times higher in the rosette and cauline leaves compared to the other tissues (Figure 1B). We then constructed *Pro_AtBBD1_:GUS* and *Pro_AtBBD2_:GUS* transgenic lines harboring the 4000-bp promoter region and 1370-bp promoter region upstream of the start codon of *AtBBD1* and *AtBBD2*, respectively, and observed the GUS expression. Consistent with the results in Figure 1B, GUS expression driven by both *AtBBD1* and *AtBBD2* promoters was detected in all the major tissues, including root, leaf, flower, and silique (Figure 1C–H). In the case of five-day-old (Figure 1C) and two-week-old (Figure 1D) plants, GUS was detected throughout all tissues in both *AtBBD1* and *AtBBD2* reporter lines. Similarly, GUS was detected throughout the cauline leaves (Figure 1E) and rosette leaves (Figure 1F) of three-week-old plants. However, *Pro_AtBBD1_:GUS* was expressed much more strongly in trichomes of the adaxial surface and the stomatal guard cells of abaxial surface, while *Pro_AtBBD2_:GUS* signal was faint in trichomes and it was expressed weakly in guard cells (the enlarged photos in Figure 1F). Moreover, in the flower, GUS was detected in almost all floral organs in both lines, but *Pro_AtBBD1_:GUS* had much stronger GUS staining than *Pro_AtBBD2_:GUS* in all floral tissues. However, no GUS signal was observed in the receptacle of *Pro_AtBBD1_:GUS* and the anther of *Pro_AtBBD2_:GUS* lines (Figure 1G). In addition, GUS was only detected in the abscission zones of the siliques of both *Pro_AtBBD1_:GUS* and *Pro_AtBBD2_:GUS* lines (Figure 1H).

Many guard cell-specific factors are well-known to modulate plant abiotic stress response, especially drought, by regulating stomatal movement. For example, MYB44, which is expressed specifically in the stomatal guard cell, represses the expression of genes encoding type 2C protein phosphatases (*PP2Cs*), the core negative regulators of ABA signal transduction, such as *ABI1*, *ABI2*, *AtPP2CA*, *HAB1*, and *HAB2*, enhancing stomatal closure, and subsequently conferring drought and salt tolerance [9]. Likewise, guard-cell specific AFBA1 (ABA-responsive FBA domain-containing protein 1) confers drought tolerance by stabilizing MYB44, and subsequently downregulating *PP2C* genes [22]. In addition, THI1 (thiamine thiazole synthase 1) positively regulates drought stress response by interacting with and repressing the kinase activity of CPK33 (Ca^2+^-dependent protein kinase 33) that reduces anion channel activity and suppresses ABA-regulated stomatal closure; both of these proteins are expressed in the guard cells [23]. Thus, the guard cell-specific expression of *AtBBD1* and *AtBBD2* indicates the possibility of their involvement in abiotic stress response in planta.

### 2.2. AtBBD1 and AtBBD2 Are Strongly Induced by ABA and Drought, and AtBBD1, but Not AtBBD2, Is also Strongly Responsive to JA

We then investigated the expression patterns of *AtBBD1* and *AtBBD2* after treating plants with ABA and JA as well as abiotic stresses, such as drought, PEG, and NaCl (Figure 2). In the mock control, we observed that the expression of *AtBBD1* was at its peak at 3 h post-treatment (hpt), at which was at noon, and then gradually decreased to its basal level at 24 hpt. On the other hand, *AtBBD2* did not show any changes in its expression during the indicated time points. Compared to their expression patterns in the mock control, the expression of *AtBBD1* and *AtBBD2* were induced by ABA within 1 hpt and reached their maximum expression at 3 and 6 hpt, respectively, and then gradually decreased. In the case of JA treatment, however, *AtBBD1* and *AtBBD2* was strongly induced at 6 and 9 hpt and then rapidly decreased. Moreover, we found that the expressions of both *AtBBD1* and *AtBBD2* were induced by drought and PEG treatments, while *AtBBD2* expression was induced by salt treatment but *AtBBD1* was not responsive to salt, suggesting that AtBBD1 is possibly involved in drought stress response.

ABA is a key plant hormone that promotes plant responses to most of abiotic stresses, including drought and salinity [24], and plays a pivotal role in regulating stomatal movement under water-deficit conditions [25]. Like ABA, JA also participates in abiotic stress responses, including drought, in addition to its well-known roles in biotic stress. The network of ABA-JA hormonal crosstalk has also emerged as playing a pivotal role in physiological responses to drought. For example, MYC2, the core regulator in JA signaling [5], was characterized as a positive regulator of ABA signaling, which binds to the promoter region *RD22* [6]. MYC2 was also reported as a positive modulator of drought tolerance through regulating *ANAC019* and *ANAC055* [7]. In our study, AtBBD1 could function primarily in drought stress response (compared to AtBBD2), and it could work in the network of ABA and JA signaling.

### 2.3. Overexpression of AtBBD1 Reduces Water Loss and Increases Proline Content, Conferring Drought Tolerance

To characterize the function of AtBBDs in drought stress, we constructed the *AtBBD1*-overexpression lines driven by the cassava vein mosaic virus (CsMV) 35S promoter and selected two homozygous lines, designated as OX5-4 and OX20-3. The expression levels of *AtBBD1* in OX5-4 and OX20-3 were around seven- and four-fold higher than the WT, respectively (Appendix A). A T-DNA insertional mutant (SALK_067060), designated as *atbbd1*, was obtained from the Arabidopsis Biological Resource Center, and an *atbbd1/atbbd2* double knockout line was obtained from Prof. Y.D. Choi at Seoul National University. Loss-of-function of both knockout lines was verified by qRT-PCR (Appendix A). We compared phenotypes of three-week-old WT, *AtBBD1-*OXs (OX5-4 and OX20-3), *atbbd1,* and *atbbd1/atbbd2* plants under drought conditions. We could not observe any phenotypic difference among these lines under normal growth conditions, while OX lines were much more tolerant to drought treatment applied by withholding water for ten days, and they recovered well after re-watering compared to the WT and KOs (Figure 3A). The survival rates of OX5-4 and OX20-3 three days after re-watering were around 84% and 76%, respectively, whereas those of the WT, *atbbd1,* and *atbbd1/atbbd2* were about 53%, 37%, and 32%, respectively (Figure 3C). In accordance with results from drought, we also observed that the OX plants showed enhanced tolerance to PEG treatment compared to the WT, with *atbbd1* and *atbbd1/atbbd2* exhibiting wilting (Figure 3B). Then, to investigate whether the drought tolerance is a consequence of reduced transpirational water loss, we measured the fresh weight of detached leaves over time and found that the transpirational water loss was lower in *AtBBD1*-OXs than the WT, while *atbbd1* and *atbbd1/atbbd2* showed the higher transcriptional water loss than the WT (Figure 3D).

Accumulation of osmoprotectants, such as proline, is well known to be associated with drought stress. Thus, we measured the proline contents of plants grown for 12 or 24 h under PEG-induced drought conditions. Consistent with results by drought treatment, the proline contents of *AtBBD1*-OX lines were much higher than that of the WT, while *atbbd1* and *atbbd1/atbbd2* showed significantly reduced proline contents than the WT (Figure 4A). Then, we examined whether *P5CS1* (*Δ1-pyrroline-5-carboxylate synthetase 1*) [26] and *PRODH* (*proline dehydrogenase*) [27], which encode the rate-limiting enzymes for proline biosynthesis and catabolism, respectively, are involved in the alteration of proline content. We found that the expression of the proline biosynthetic gene *P5CS1* increased over two-fold in *AtBBD1*-OX plants, but decreased about three-fold in *atbbd1* and *atbbd1/atbbd2* relative to the WT under the PEG-mediated drought stress (Figure 4B). By contrast, the expression of the proline catabolic gene *PRODH* was down-regulated by about half in the *AtBBD1*-OX plants compared to the KOs at the early point of drought stress (Figure 4C).

AtBBD1 and AtBBD2 have a very high sequence identity, and they function as nucleases [21]. *AtBBD2* showed a very similar expression pattern to *AtBBD1*, despite its generally lower expression than AtBBD1, but the double knock-out (*atbbd1/atbbd2*) appeared to have no differences in drought phenotype, water loss, proline content accumulation, or proline-related gene expression compared to the single knock-out mutant *atbbd1*. Taken together, our findings clearly suggest that overexpression of *AtBBD1* confers drought tolerance by reducing transpirational water loss and increasing proline content. Moreover, *AtBBD1* alone is sufficient to trigger drought tolerance in planta, and *AtBBD1* and *AtBBD2* are perhaps not functionally redundant, at least in drought stress response.

### 2.4. AtBBD1 Enhances the ABA-Mediated Stomatal Closure

OmBBD, the wild rice ortholog of AtBBD1, is a cofactor-independent novel bifunctional nuclease that is involved in ABA-mediated priming of callose reinforcement and that enhances JA-mediated defense responses against *Botrytis cinerea* [17]. Because we recently reported that Arabidopsis AtBBD1 and AtBBD2 proteins possess similar DNase and RNase activities as OmBBD [21], and because we found that AtBBD1 is strongly responsive to ABA and JA (Figure 2), we postulated that AtBBD1 has an ABA and/or JA-mediated regulatory role in planta.

To verify whether AtBBD1 has a possible role in ABA-mediated responses, we examined the rate of seed germination (as radicle emergence) and the percentage of emergence of green cotyledons after treating them with different concentrations (0, 1, and 1.5 µM) of ABA. On MS medium without ABA, the germination rates of all seeds were almost 100%. By increasing ABA concentration, however, the germination rate of the WT seeds significantly dropped, as expected, and the two OX lines had much more reduced germination rates compared to the WT, while *atbbd1* and *atbbd1/atbbd2* lines showed increased germination rates compared to the WT (Figure 5A). Consistently, the number of seedlings with green cotyledons remarkably decreased by increasing ABA concentrations. In the case of 1.5 µM ABA treatment, the initial 100% emergence of green cotyledons decreased to about 20% in the WT and 7.5% and 9.3%, respectively, in OX5-4 and OX20-3, whereas it was higher in *atbbd1* at 28.1% and in *atbbd1/atbbd2* at 31.7% (Figure 5B,C). However, as expected, there was no significant difference between *atbbd1* and *atbbd1/atbbd2* in seed germination and emergence of green cotyledons. To explore the role of AtBBD1 in drought stress-induced ABA signaling, stomatal aperture was measured in the presence or absence of ABA. There was no difference in stomatal apertures of all the lines in the absence of ABA, but ABA treatment led to an increased stomatal closure in the OX lines, such 1.3-fold in OX5-4 and 1.2-fold in OX20-3, compared to the WT. In contrast, the stomatal apertures of *atbbd1* and *atbbd1/atbbd2* were about 1.2-fold wider than that of the WT in the presence of ABA (Figure 5D,E). These data demonstrate that AtBBD1 enhances stomatal closure in an ABA-dependent manner.

Since drought response is regulated not only by ABA, but also by JA and ABA-JA synergistic action, we investigated whether *AtBBD1* is regulated by JA-mediated signaling. Previous work has shown that MYC2-mediated JA signaling inhibits the primary root growth of Arabidopsis seedlings [28]. Therefore, we compared the JA sensitivity of primary roots of the WT, overexpression lines (OX5-4 and OX20-3) and mutants (*atbbd1* and *atbbd1/atbbd2*). However, we could not find any differences in JA-mediated primary root growth retardation among them, indicating that JA signaling is not directly affected by the alteration of *AtBBD1* expression (Appendix A). A recent FRET kinase reporter system revealed that ABA alone can regulate the activation of *SnRK2/OST1* and stomatal closure, but MeJA alone does not, demonstrating that ABA-JA synergistic interaction is not essential for ABA-mediated stomatal movement [29]. Therefore, we then examined whether the expression of *MYC2* remained unchanged in the overexpression lines and mutants when compared with WT after drought stress. MYC2 is regarded as a hub for ABA-JA hormonal crosstalk, and acts as a positive regulator of drought stress response [6]. Our qRT-PCR analysis showed that two independent *AtBBD1*-OX lines had significant down-regulation of *MYC2* under drought stress (Appendix A), opposite of their drought tolerance phenotypes. To the best of our knowledge, there has been no report that the negative regulation of *MYC2* can trigger drought tolerance. Therefore, down-regulation of *MYC2* by AtBBD1 while regulating ABA-JA crosstalk by drought stress response provides a new insight into ABA-JA interactive pathways, but further experiments are necessary to fully unravel the puzzle.

### 2.5. Overexpression of AtBBD1 Enhances the Expression of Key Regulatory Genes in ABA-Dependent Pathway

To gain insight into the mechanism of AtBBD1 in drought stress response, we performed RNA sequencing (RNA-seq) using the WT, OX5-4, and *atbbd1* plants treated with 20% PEG for 6 h, including the WT without treatment as a control. We generated a heatmap of RNA-seq expression z-scores computed for all genes that were differentially expressed (Appendix A). By applying specific filters to screen genes with at least two-fold difference in the expression level, we identified 388 DEGs that were up-regulated in OX5-4 and were down-regulated in *atbbd1* (Appendix A). Using Gene Ontology (GO) analysis, we found that most of the DEGs belong to the groups related to ABA or drought, such as response to ABA, response to drought, response to osmotic stress, ABA activated signaling, and negative regulation of ABA signaling, etc. (Appendix A).

Because *AtBBD1* is regulated by ABA signaling (Figure 5A–E) and because the ABA-dependent signaling cascade is a major player in drought response, we identified the expression patterns of some key genes related to ABA and its downstream drought stress signaling cascade using RNA-seq data (Figure 6A,B). As shown in Figure 6B, the expression of genes related to ABA biosynthesis, Snf-1-related protein kinases, and drought stress response increased in OX5-4 plants, but decreased in *atbbd1* plants, compared to the WT, after 6 h of drought treatment, consistent with our results on drought phenotypes.

Since we analyzed the RNA-seq analysis in a single sequencing run, we validated the expression profiles of some key genes involved in ABA-dependent drought stress response by qRT-PCR after drought treatment (Figure 7). Consistent with our RNA-seq data, the expression of *ABA1* (*ABA deficient 1*) [30] and *NCED3* (*nine-cis-epoxycarotenoid dioxygenase 3*) [2], two genes related to ABA biosynthesis, was higher in OX plants than in the WT, whereas it greatly decreased in *atbbd1* and *atbbd1/atbbd2* compared to the WT at both 6 and 12 h after drought treatment. It is well-known that the induction of ABA biosynthesis-related genes, including *ABA1* and *NCED3*, under stress conditions results in the enhanced production of ABA in various plant tissues, leading to tolerance to stresses [2,30]. Similarly, the expression of six well-characterized drought stress-responsive marker genes, *MYB44*, *NAC019* (*NAC domain containing protein 19*), *RD29B* (*responsive to desiccation 29B*), *RAB18* (*responsive to ABA 18*), *RD20* (*responsive to desiccation 20*), and *RD22* (*responsive to desiccation 22*), was also highly induced in OX plants than the WT, while it was greatly reduced in *atbbd1* and *atbbd1/atbbd2* compared to the WT after 6 h and 12 h of drought treatment (Figure 7).

Since it is known that drought response in Arabidopsis is considerably regulated by ABA-independent signaling pathways, we analyzed our RNA-seq data in-depth to unearth whether an ABA-independent signaling pathway is also involved in AtBBD1′s drought tolerance regulation. We could not find any significant altered expression of *DREB2A* and *DREB2B,* two well-studied genes for ABA-independent drought stress signaling (Appendix A). However, *ERD1*, a gene also related to the ABA independent pathway, showed increased expression in OX plants, whereas it had decreased expression in *atbbd1* plants compared to the WT (Appendix A). *ERD1* expression can be regulated by ABA and drought inducible NAC TFs, NAC019, and NAC055, which have been reported to function via ABA-dependent and ABA-JA synergistic pathways during drought stress [31].

We also noted that the expression of genes related to ABA catabolism and PP2Cs, which are negative regulators of ABA signaling, increased in OX5-4 and decreased in *atbbd1* compared to the WT after 6 h of drought treatment (Figure 6B). Consistent with the RNA-seq data, our qRT-PCR analyses also revealed the expression of *CYP707A3* (*Cytochrome 707A3*) and *PP2CA* was increased in OX plants, but was decreased in single and double knock-out plants compared to the WT at the indicated time points of drought treatment (Figure 7). This result could be explained by the AtBBD1-derived positive feedback, which may increase the endogenous ABA level, contributing to the higher expression of those genes [32,33]. The involvement of JA signaling for the function of *AtBBD1* during drought stress is further confirmed by our RNA-seq data, as a few genes related to JA-signaling appeared to be enriched (Appendix A). Taken together, our results imply that the drought tolerance conferred by AtBBD1 is regulated by ABA-dependent drought response signaling.

### 2.6. Overexpression of AtBBD1 Increases the Expression of Drought Stress-Responsive Genes

To identify other regulatory genes responsible for the drought tolerance phenotype of *AtBBD1*-OX plants, we generated a heatmap of DEGs related to the response to drought and osmotic stress, and found that the expression of a number of well-known positive regulators in drought response, such as *MYB2* [6], STZ/*ZAT10* (*salt tolerance zinc finger*) [34], *LEA4-5* (*late embryogenesis abundant 14*) [35], *RPK1* (*receptor-like protein kinase 1*) [36], and *LTP3* (*lipid transfer protein 3*) [37], etc., was enhanced in the OX5-4 line but was reduced in *atbbd1* compared to the WT (Figure 8A). We performed qRT-PCR for three genes, *LTP3, LEA4-5,* and *ZAT10*, and observed that the expression of these genes was greatly increased in OX plants and was sharply decreased in *atbbd1* and *atbbd1/atbbd2* compared to the WT plants after 6 h of drought treatment (Figure 8B), which is consistent with the RNA-seq data (Figure 8A). These data imply that our single sequencing run RNA-seq data is quite reliable and suggests that the drought tolerance and susceptible phenotype of *AtBBD1*-OX and *atbbd1* plants, respectively, are associated with the altered expression of well-known drought stress responsive genes.

We also identified significant GO enrichment related to cold (Appendix A) and differential expression of some cold-stress responsive genes like *KIN1* and *RAB18* (Figure 7 and Figure 8), indicating that *AtBBD1* possibly has an important role in cold response in Arabidopsis. Further experiments are required to elucidate the physiological role of *AtBBD1* in cold stress. We recently reported that AtBBD1 can cleave both DNAs and RNAs, including mRNA, in vitro [21] We are not sure whether the enhanced expression of key genes is affected by its nuclease activity. Although there are recent reports that RNA processing, such as disruption of alternative splicing, mimics abiotic stress response through activating ABA signaling [38], no nuclease has been reported to be involved in ABA-mediated drought resistance. Thus, future research is needed to address whether nuclease activity is crucial for ABA-mediated drought resistance by AtBBD1.

## 3. Materials and Methods

### 3.1. Plant Materials

Seeds of the wild-type (WT, *Arabidopsis thaliana* ecotype Columbia) and all transgenic plants were surface sterilized with 70% ethanol for 1 min followed by treatment with 2% sodium hydroxide for 5 min and then washed five times with sterile water. All seeds were grown in soil (Sunshine Mix#5, Sun Gro Horticulture) or on 1/2 strength Murashige and Skoog (MS) agar media [39] supplemented with 1% (*w*/*v*) sucrose and 1% (*w*/*v*) agar, at 23 ± 2 °C and 50% relative humidity under 16 h-light/8 h-dark conditions. The T-DNA insertion mutant *atbbd1* (SALK_067060) was obtained from the Arabidopsis Biological Resource Centre (http://www.arabidopsis.org/ (accessed on 17 February 2021)), and the *atbbd1/atbbd2* double knock-out line was provided by Prof. Yang Do Choi in Seoul National University (Appendix A).

To analyze the expression patterns of *AtBBD1* and *AtBBD2* in response to ABA and JA, two-week-old Arabidopsis (Col-0) plants grown on 1/2 MS solid media were treated with 100 µM ABA or 100 µM JA. For dehydration treatment, the plants were placed in a laminar flow hood without the plate lid. For treatment of PEG and salt, three-week-old soil-grown plants were watered with 20% PEG 6000 or 200 mM NaCl. Then, all the samples were collected 0, 1, 3, 6, 9, 12, and 24 h after treatment to check whether expression of genes was affected by the circadian rhythm.

### 3.2. Construction of Transgenic Plants

To construct *Pro_BBD1_:GUS* and *Pro_BBD2_:GUS* transgenic plants, the 4000-bp and 1370-bp putative promoter regions upstream from the start codon of *AtBBD1* and *AtBBD2* genes, respectively, were amplified from Arabidopsis genomic DNA by PCR using the primers listed in the Appendix A, and were inserted into the corresponding *BamHI* site of the pBI101 vector (Clontech). For *AtBBD1*-overexpressing transgenic Arabidopsis plants, the coding region of *AtBBD1* (AT1g75380) was amplified with the primers listed in Appendix A using PrimeSTAR™ HS DNA Polymerase (Takara Bio), digested with *XbaI* and *BamHI*, and inserted into the corresponding site of the binary plant transformation vector pCsVMV1300 [40].

The vector constructs were transformed into *Agrobacterium tumefaciens* strain GV3101 using the freeze-thaw method [41]. Transgenic plants generated using the floral dipping method [42] were grown on a 1/2 MS solid selection medium containing 250 mg L^−1^ carbenicillin and 30 mg L^−1^ hygromycin or 50 mg L^−1^ kanamycin for *AtBBD1*-overexpressing or *Pro_BBDs_:GUS* plants, respectively. The surviving plants were then transferred to soil. For *AtBBD1*-overexpression lines, two independent T_3_ homozygous lines (OX5-4 and OX20-3) were selected and used in this study.

### 3.3. Subcellular Localization Analysis

To generate *Pro_35S_:BBD1-sGFP* and *Pro_35S_:BBD2-sGFP* constructs, full-length cDNAs of *AtBBD1* and *AtBBD2* were amplified, digested with *Xba*I and *Bam*HI, and ligated in-frame to the C terminus of the green fluorescence protein (GFP) gene of the vector 326-sGFP [43]. The constructs were introduced into Arabidopsis protoplasts together with *Pro_35S_:NLS-RFP*, a chimeric red fluorescent protein (RFP) construct containing a nuclear localization signal, using a PEG-mediated transformation method with minor modifications [44], and then incubated overnight under dim light. To analyze the subcellular localization, the GFP or RFP signal in each sample was captured by a confocal laser scanning microscope (LSM 510 META, Zeiss). An argon laser (488 nm of excitation) was set with 505–530 band-pass for GFP, while a HeNe laser (543 nm) was set with 560–615 bandpass for RFP detection. LSM image browser software (Zeiss) was used to adjust the images.

### 3.4. GUS Histochemical Staining Analysis

For GUS histochemical staining, samples were incubated with X-Gluc solution (1 mM 5-bromo-4-chloro-3-indolyl-β-D-glucuronic acid, 10 mM Na_2_EDTA, 0.5 mM potassium ferrocyanide, 0.5 mM potassium ferricyanide, 0.1% (*v*/*v*) Triton X-100, 50 mM NaPO_4_ buffer, pH 7.0) at 37 °C overnight. After staining, the chlorophyll was cleared by immersing the samples in 70% EtOH and 100% EtOH each for one day. The tissues were observed with an optical microscope (Olympus BX-51).

### 3.5. RNA Isolation and qPCR Analysis

Total RNA was isolated using RNAiso (Takara Bio) according to the manufacturer’s instructions. Three µg of Arabidopsis total RNA was reverse transcribed using RevertAid First Strand cDNA Synthesis Kit (Thermo Fisher Scientific) according to the manufacturer’s protocol. Quantitative real time PCR (qRT-PCR) was performed using the KAPA SYBR^®^ FAST qPCR kit (Kapa Biosystems) in a LightCycler^®^480 II (Roche). Ten ng of cDNA were amplified in a 20-μL reaction using the following conditions: denaturation at 95 °C for 5 min and 45 cycles of 95 °C for 10 s, 57 °C for 10 s, and 72 °C for 20 s. The relative expression was calculated by the comparative Ct method (threshold cycle number at the cross-point between amplification plot and threshold) as determined by analysis with LightCycler^®^480 II software. *PP2CAA3* (AT1g13320) or *eIF4a1* was used as the internal reference to normalize the values. Each cDNA sample was amplified in triplicate to circumvent the experimental error. Three independent experiments were carried out for biological replication. All primers used for qRT-PCR are shown in Appendix A.

### 3.6. Drought Treatment and Water Loss

For drought treatment, three-week-old soil-grown Arabidopsis plants were subjected to a dehydration treatment by withholding watering for 10 days. Drought tolerance was assessed three days after re-watering and the survival rate was calculated by counting the number of plants whose growth resumed. To measure the percentage of transpiration water loss, 20 rosette leaves from three-week-old soil-grown plants were detached, weighed, and placed with the abaxial side facing downwards on a filter paper at room temperature, and the leaves were weighed again after 1, 2, 3 and 4 h. The water loss was calculated as the percentage of initial fresh weight. For PEG treatment, three-week-old soil-grown Arabidopsis plants were watered with 20% PEG 6000, and the wilting phenotype was observed three days after treatment. These experiments were carried out with three biological replications.

### 3.7. Quantification of Proline Contents

Proline content was measured as described by Bates et al. (1973) with minor modifications. Leaves of three-week-old plants were freeze-dried, homogenized in 3% sulfosalicylic acid, and then centrifuged at 13,000× *g* for 20 min. The supernatants were transferred into new tubes and equal volumes of acetic acid and ninhydrin reagent (1.25 g ninhydrin dissolved by warming in 30 mL glacial acetic acid and 20 mL 6 M phosphoric acid) were added. The tubes were boiled for 60 min and then kept on ice for 30 min. An equal volume of toluene was added to each sample, mixed vigorously and centrifuged at 1000× *g* for 5 min at room temperature. The absorbance of the upper layer was measured at 520 nm using a Synergy^TM^ H1 Hybrid Multi-Mode Reader (BioTek^®^ Instruments, Inc.).

### 3.8. Seed Germination Assays

Seeds of the WT, *atbbd1*, *atbbd1/atbbd2*, and *AtBBD1*-Ox lines (OX5-4 and OX20-3) were stratified at 4 °C for two days in the dark prior to germination. A total of 100 seeds of each line were plated on MS medium supplemented with 1.0 and 1.5 µM ABA. Seeds were considered to have germinated when the radicle protruded through the seed coat. All seeds were scored daily for five days. The germination rate was calculated as a percentage of the total number of seeds plated. The emergence of green cotyledons was counted on the seventh day after plating. The percentage of green cotyledon emergence was calculated as a percentage of the total number of seeds plated. These two experiments were carried out in triplicate.

### 3.9. Measurement of Stomatal Aperture

Ten rosette leaves from five three-week-old plants were floated in stomatal opening buffer (50 mM KCl and 10 mM MES-KOH, pH 6.15, 10 mM CaCl_2_) under light conditions for 2 h. Stomatal closure was induced by replacing the buffer with fresh buffer containing 10 µM ABA and by incubating the leaves for an additional 2 h. Stomatal apertures in epidermal peels were observed under a microscope (model BX-51; Olympus). Thirty stomata per leaf were calculated using ImageJ software (https://imagej.nih.gov/ij/ (accessed on 17 February 2021)). The buffer without ABA was used as a control. The experiment was carried out in three biological replications.

### 3.10. RNA Sequencing

RNA-sequencing was performed using total RNAs extracted from three-week-old leaves of the WT, OX5-4, and *atbbd1* lines grown under drought conditions (20% PEG, 6 h), because *AtBBD1* gene expression is highest under this condition. Total RNAs extracted from at least five leaves from five individual plants of each genotype were treated with RNase-free DNase I (Invitogen). The concentration of RNA was determined using a DropSense 96 (PerkinElmer) and RiboGreen (Invitogen), and the integrity of RNA was analyzed with a Bioanalyzer RNA Chip (Agilent Technologies). The RNA with a RIN number ≥ 6 was considered for RNA sequencing. The sequencing library was constructed using TruSeq Stranded mRNA Library Prep (Illumina) using 1 μg of total RNA of each sample. The libraries were further validated by Agilent 2100 Bioanalyzer (Agilent Technologies). Afterward, sequencing was performed on a Nextseq 500 Sequencer System (Illumina).

### 3.11. Data Access

The RNA-seq data of this study have been deposited in NCBI Gene Expression Omnibus (GEO, http://www.ncbi.nlm.nih.gov/geo/ (accessed on 17 February 2021)) with accession number GSE152163.

### 3.12. Transcriptome Data Analysis

The raw sequence reads were processed for removal of adapter sequences, followed by qualitative analysis of raw reads using FastQC (http://www.bioinformatics.babraham.ac.uk/projects/fastqc (accessed on 17 February 2021)). The high-quality reads were then mapped against the *Arabidopsis thaliana* reference genome TAIR10 (downloaded from http://arabidopsis.org (accessed on 17 February 2021)) using HISAT2 version 2.1.0 [45] with default parameters. Differentially expressed genes were identified using the Cufflinks pipeline [46] and were filtered based on fold-change. Genes with at least two-fold change of RPKM values (reads per kilobase of transcript, per million mapped reads) between two datasets were considered as differentially expressed genes (DEGs). Gene ontology (GO) enrichment analysis was performed using AgriGo Version 2 [47]. We selected the GO terms with Benjamini adjusted *p*-value ≥ 0.05. The heatmap was generated using R programming language (version 4.0.0) and “gplots” package (version 3.0.3).

### 3.13. Statistical Analysis

All statistical comparisons between variants were determined using Graphpad Prism 7 statistical software. Statistical analysis was assessed using one-way ANOVA with post-hoc Tukey test with a 0.05 level of significance (95% confidence interval).

## Figures and Tables

**Figure 1 ijms-22-02936-f001:**
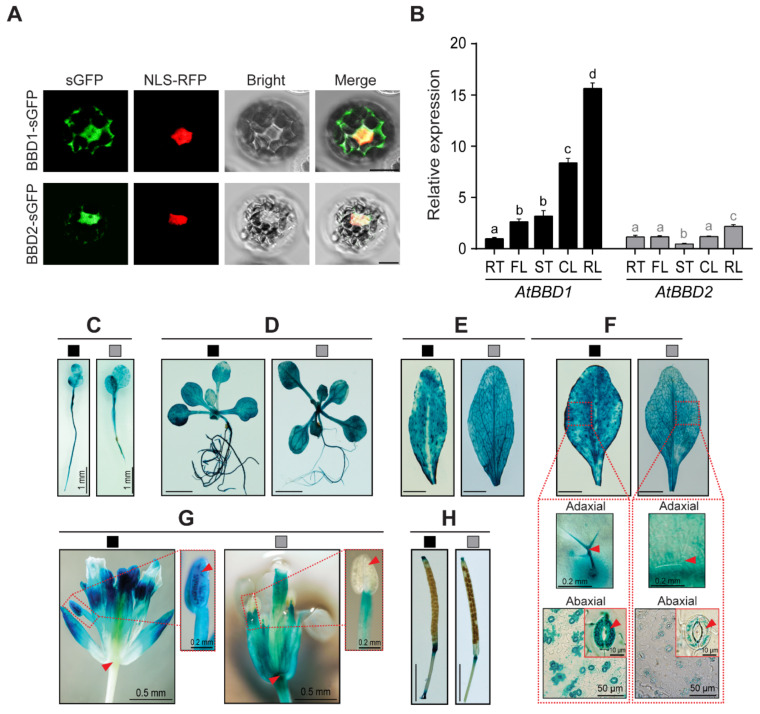
Subcellular localization and tissue-specific expression of *AtBBD1* and *AtBBD2.* (**A**) Subcellular localization of BBD1 and BBD2. *Pro_35S_:AtBBD1-sGFP* or *Pro_35S_:AtBBD2-sGFP* along with *Pro_35S_:NLS-RFP* were transformed into protoplasts isolated from two-week-old Arabidopsis seedlings using a PEG-mediated transformation method. Images of the transformed protoplasts were captured by a confocal laser scanning microscope (LSM 510 META, Zeiss) 20 h after transformation. Scale bar = 10 µM. (**B**) Spatial expression of *AtBBD1* and *AtBBD2* determined by qRT-PCR. The relative gene expression was calculated and normalized to the reference gene *PP2AA3.* The normalized mRNA level in root of *AtBBD1* was set to 1. Total RNAs were obtained from the rosette leaves (RL), cauline leaves (CL), stem (ST), flower (FL), and root (RT). Data represent the mean ± standard error values of three independent experiments. One-way ANOVA with post-hoc Tukey test was used for the statistical comparison of all tissues. Different letters indicate significant differences among tissues for each gene (*p* < 0.05). (**C**–**H**) Promoter activities of *AtBBD1* and *AtBBD2* genes across developmental stages were demonstrated by histochemical GUS analysis of transgenic *ProBBD1:GUS* and *ProBBD2:GUS* lines. In each panel, black and gray squares represent *AtBBD1* and *AtBBD2*, respectively. Red arrow indicates the differential GUS staining between *AtBBD1* and *AtBBD2.* Bar = 10 mm unless written in the figure. (**C**) Five-day-old seedling. (**D**) Two-week-old plants. (**E**) Cauline leaves from three-week-old plants. (**F**) Rosette leaves from three-week-old plants. (**G**) Solitary flower. (**H**) Siliques from five-week-old plants.

**Figure 2 ijms-22-02936-f002:**
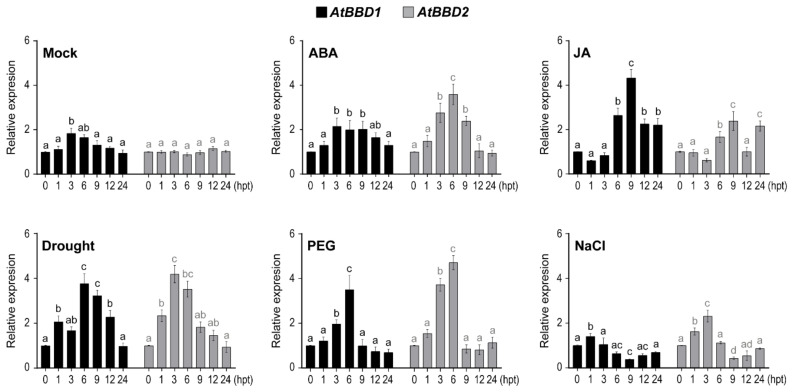
Expression of *AtBBD1* and *AtBBD2* genes in Arabidopsis at various times in response to abiotic stresses. The gene expression was analyzed by qRT-PCR. Two-week-old Arabidopsis (Col-0) plants grown on 1/2 MS agar media were treated with 100 µM abscisic acid (ABA) or 100 µM jasmonic acid (JA) or kept in a lamina flow hood for the dry condition, or transferred to 1/2 MS liquid media containing 20% PEG 6000 (PEG), or were transferred to 1/2 MS liquid media containing 200 mM NaCl. The gene expression was calculated and normalized to the reference gene *eIF4a1*. The normalized mRNA levels in mock-treated wild-type plants at 0 h were set to 1. The relative gene expression values in response to treatments were normalized to mock for each time point. Data represent the mean ± standard error values of three independent experiments. One-way ANOVA with post-hoc Tukey test was used for the statistical comparison of all time points. Different letters indicate significant differences among times for each gene after treatment (*p* < 0.05).

**Figure 3 ijms-22-02936-f003:**
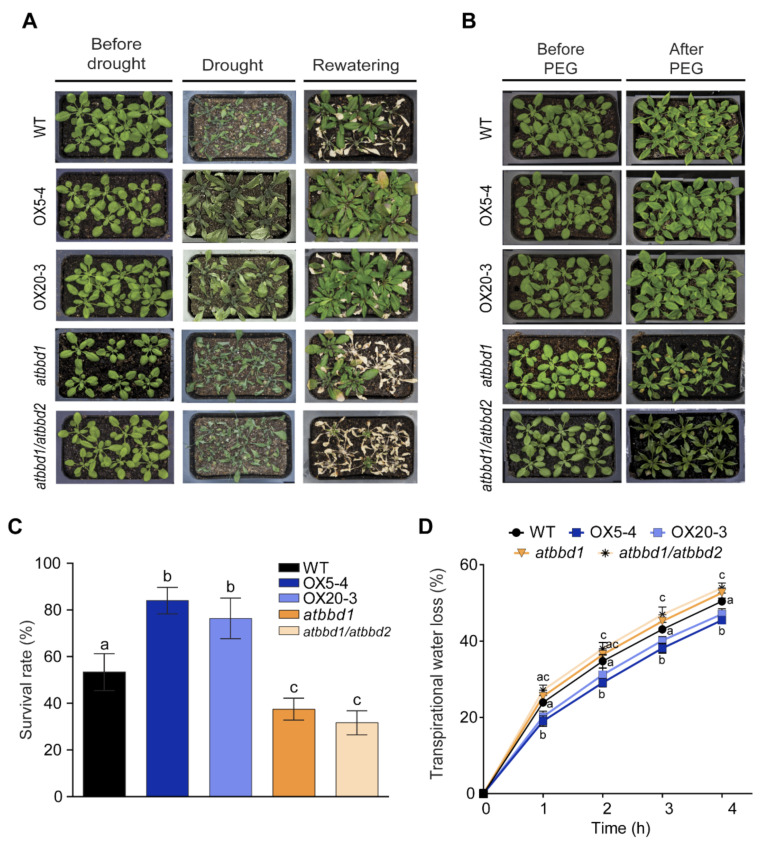
Drought stress response of the WT, two *BBD1*-Ox, *atbbd1*, and *atbbd1/atbbd2* plants. (**A**) Response of the WT, OX5-4, OX20-3, *atbbd1,* and *atbbd1/atbbd2* plants under drought stress conditions. Three-week-old plants were subjected to drought stress by withholding water for 12 days, followed by rewatering for three days. (**B**) Response of the WT, OX5-4, OX20-3, *atbbd1,* and *atbbd1/atbbd2* plants under PEG treatment. Three-week-old plants were stressed by watering with 20% PEG 6000 treatment for three days, and wilting phenotypes were observed. (**C**) Survival rates of plants after rewatering. Data represent the mean ± standard error of three independent experiments, each evaluating 32 plants. (**D**) Transpirational water loss from the leaves of the WT, OX5-4, OX20-3, *atbbd1,* and *atbbd1/atbbd2* plants at various time points after detachment of leaves. Data represent the mean ± standard error of three independent experiments, each evaluating 30 leaves. One-way ANOVA with post-hoc Tukey test was used for the statistical comparison of all genotypes. Different letters indicate significant differences among genotypes (*p* < 0.05).

**Figure 4 ijms-22-02936-f004:**
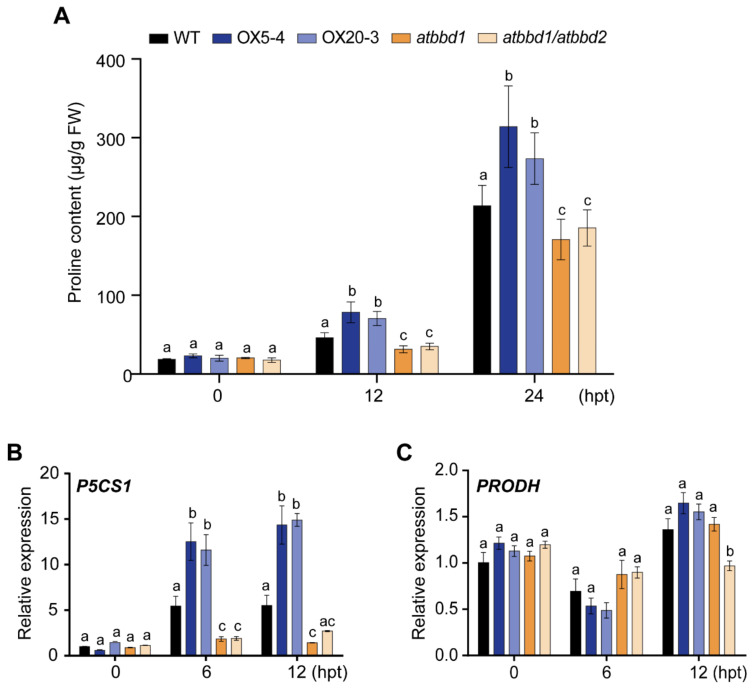
Analysis of proline in the WT, two *BBD1*-Ox lines, *atbbd1,* and *atbbd1/atbbd2* plants under drought stress conditions. (**A**) Free proline contents were measured at designated time points from the leaves of three-week-old soil-grown plants stressed by watering with 20% PEG 6000. (**B**) The expression of the proline biosynthesis gene *P5CS1* was analyzed by qRT-PCR. (**C**) The expression of the proline catabolic gene *PRODH* was analyzed with qRT-PCR. The RNA was isolated at designated time points from three-week-old soil-grown plants stressed by watering with 20% PEG 6000. The relative gene expression was calculated and normalized to the reference gene *eIF4a1*. The normalized mRNA levels at 0 h were set to 1. Data represent the mean ± standard error values of three independent experiments. One-way ANOVA with post-hoc Tukey test was used for the statistical comparison of all genotypes. Different letters indicate significant differences among genotypes at each time point (*p* < 0.05).

**Figure 5 ijms-22-02936-f005:**
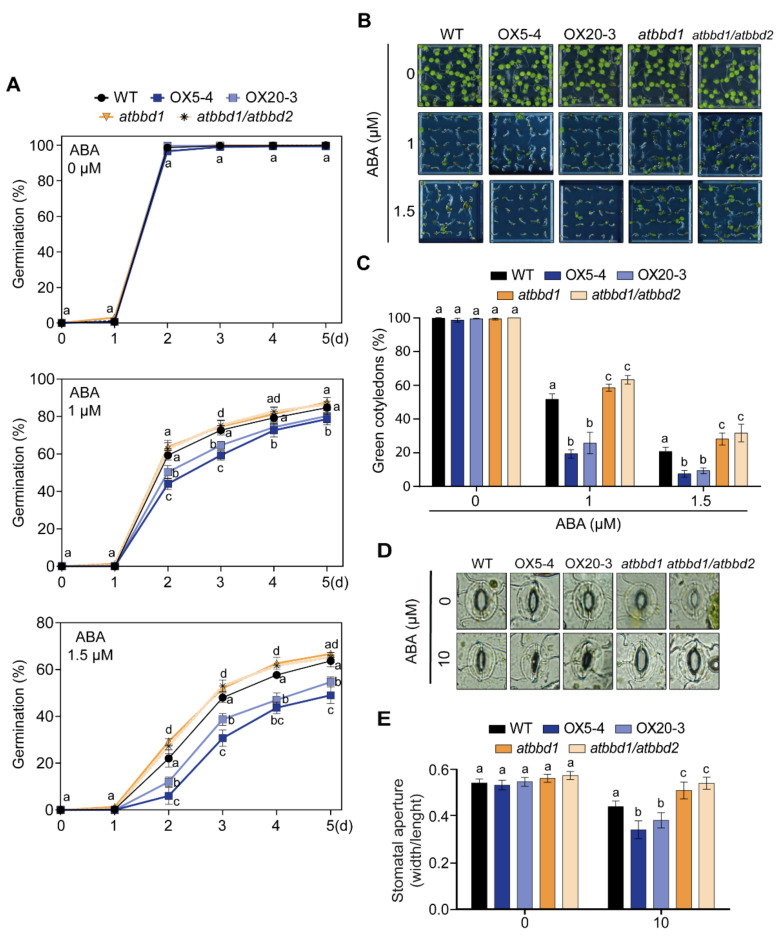
ABA sensitivity of the WT, two *BBD1*-Ox lines, *atbbd1*, and *atbbd1/atbbd2* plants during germination. (**A**) Seed germination of the WT, OX5-4, OX20-3, *atbbd1,* and *atbbd1/atbbd2* plants in response to ABA. Seeds were germinated on 1/2 MS agar plates containing 0.0 or 1 or 1.5 μM ABA. The seed germination rates were calculated at the designated times over five days. Data represent the mean ± standard error values of three independent experiments (*n* = 100). (**B**) Growth of the WT, OX5-4, OX20-3, *atbbd1,* and *atbbd1/atbbd2* plants on 1/2 MS agar plates containing the indicated concentrations of ABA. Representative photographs were taken seven days after plating. (**C**) Quantification of green cotyledons in the WT, OX5-4, OX20-3, *atbbd1,* and *atbbd1/atbbd2* plants was performed seven days after plating. Data represent the mean ± standard error values of three independent experiments (*n* = 100). (**D**) Stomatal apertures in the WT, OX5-4, OX20-3, *atbbd1,* and *atbbd1/atbbd2* plants treated with 0 and 10 μM ABA. Leaf peels were harvested from three-week-old plants and stomatal apertures were measured under a microscope. (**E**) Quantification of stomatal aperture. Data represent the mean ± standard error of three independent experiments, each evaluating 10 leaves from five independent plants (*n* = 300). One-way ANOVA with post-hoc Tukey test was used for the statistical comparison of all genotypes. Different letters indicate significant differences among genotypes after treatment (*p* < 0.05).

**Figure 6 ijms-22-02936-f006:**
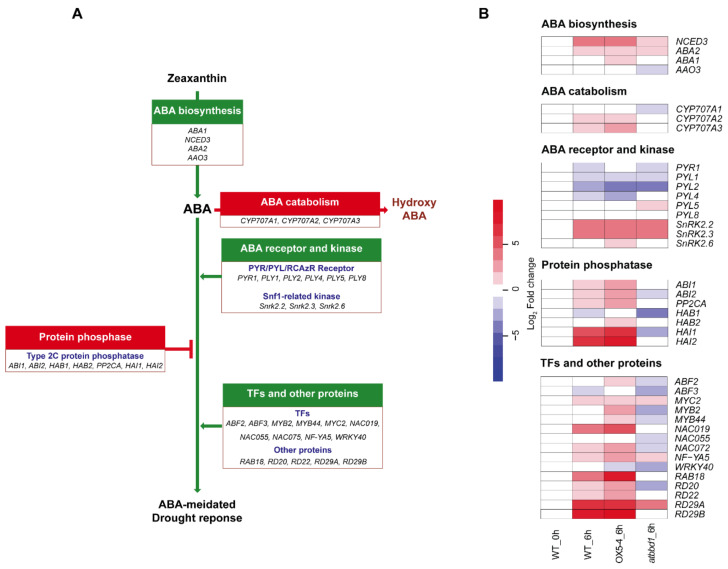
Expression of genes related to ABA and drought stress signaling cascades. (**A**) ABA-dependent signaling network of 38 key genes involved in drought stress response. Genes in the green boxes are positive regulators, while genes in the red boxes are negative regulators of ABA signaling. (**B**) Heatmap showing the expression fold change of 38 key genes involved in the ABA-dependent drought stress signaling network in *Arabidopsis thaliana* as determined by RNA-seq in the WT, OX5-4, and *atbbd1* plants under drought stress conditions. The relative fold change was calculated and normalized with the RPKM value of WT plants without drought treatment. The RPKM value of WT at 0 h was set to 1.

**Figure 7 ijms-22-02936-f007:**
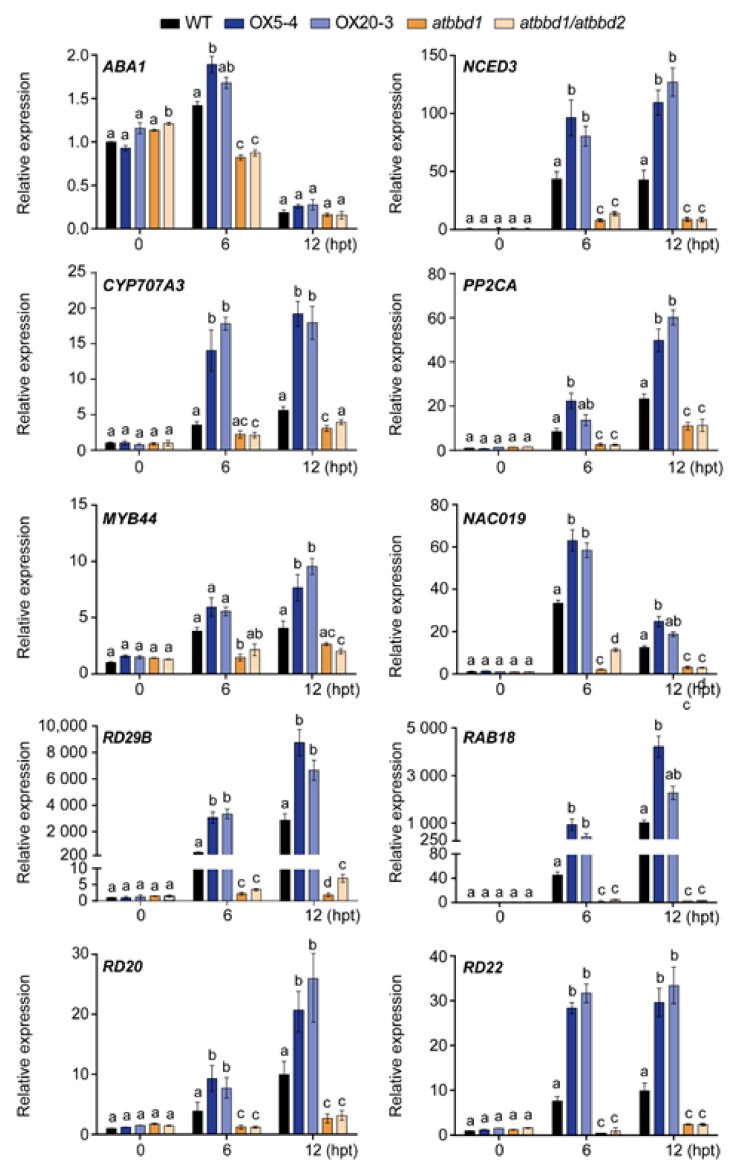
Expression of ABA-inducible and drought stress-inducible marker genes in the WT, OX5-4, OX20-3, *atbbd1,* and *atbbd1/atbbd2* lines under drought stress conditions, as assayed by qRT-PCR. Three-week-old soil-grown plants were stressed by watering with 20% PEG 6000, and gene expression was analyzed at designated time points. Relative expression levels of *ABA1*, *NCED3*, *CYP707A3*, *PP2CA*, *MYB44*, *NAC019*, *RD29B*, *RAB18*, *RD20,* and *RD22* in the WT, OX5-4, and *atbbd1* plants. The relative gene expression was calculated and normalized to the reference gene *eIF4a1.* The normalized mRNA levels at 0 h were set to 1. Data represent the mean ± standard error values of three independent experiments. One-way ANOVA with post-hoc Tukey test was used for the statistical comparison of all genotypes. Different letters indicate significant differences among genotypes at each time point (*p* < 0.05).

**Figure 8 ijms-22-02936-f008:**
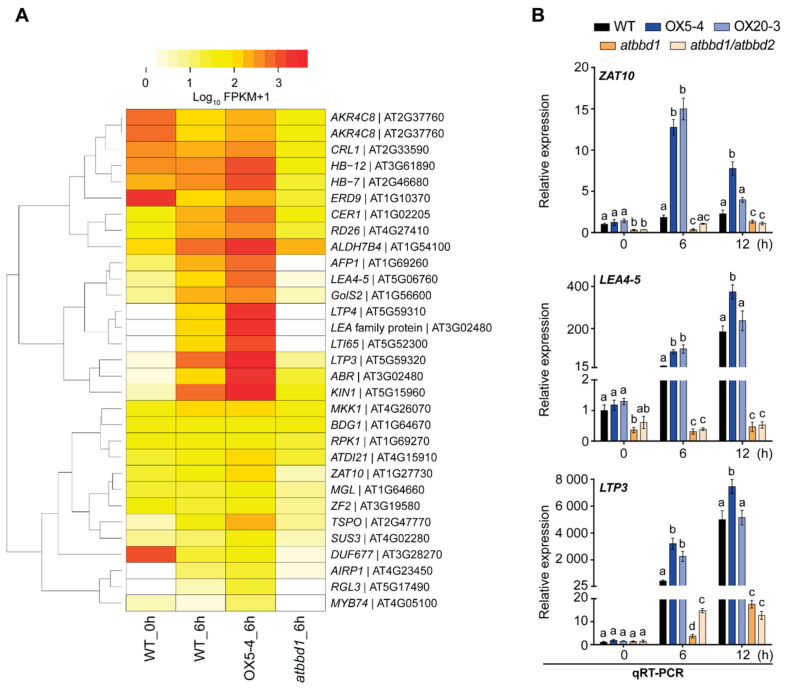
Expression of genes related to drought stress that appeared in RNA-seq data in the WT, OX5-4 and *atbbd1* plants. Three-week-old soil-grown plants were stressed by watering with 20% PEG 6000 and gene expression was analyzed at designated time points. (**A**) Heatmap showing the expression (RPKM) of 31 drought responsive genes that are possibly regulated by *AtBBD1*. (**B**) Relative expression levels determined by qRT-PCR of *ZAT10*, *LEA4-5,* and *LTP3* in the WT, OX5-4, and *atbbd1* plants. The relative gene expression was calculated and normalized to the reference gene *eIF4a1*. The normalized mRNA levels at 0 h were set to 1. Data represent the mean ± standard error values of three independent experiments. One-way ANOVA with post-hoc Tukey test was used for the statistical comparison of all genotypes. Different letters indicate significant differences among genotypes at each time point (*p* < 0.05).

## Data Availability

The RNA-seq data of this study is available in NCBI Gene Expression Omnibus (GEO, http://www.ncbi.nlm.nih.gov/geo/ (accessed on 17 February 2021)) with accession number GSE152163.

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
