# Peer review of "Overexpression of AtBBD1, Arabidopsis Bifunctional Nuclease, Confers Drought Tolerance by Enhancing the Expression of Regulatory Genes in ABA-Mediated Drought Stress Signaling"

_ijms, 2021, doi:10.3390/ijms22062936_

Round 1

Reviewer 1 Report

This manuscript is describing the characterization of the AtBBD1 and AtBBD2 genes from the evidence that overexpression of AtBBD1 confers drought tolerance by enhancing the expression of regulatory genes in plant hormone abscisic acid (ABA)-mediated drought stress signaling in Arabidopsis. This manuscript also describesthe identification of key regulatory genes in ABA-dependent pathway by using RNA-seq analysis in Arabidopsis plants overexpressing AtBBD1. This manuscript might be accepted for publication in International Journal of Molecular Sciences after the following points would be addressed.

1. In Figure 2, the authors observed that the expression of AtBBD1 was at its peak at 3 h post treatment in the case of mock control. Therefore, it would be desirable to normalize the relative expression levels of both AtBBD1 and AtBBD2 genes by the levels of mock as a control.

2. In Figure 7, two genes related to ABA biosynthesis, ABA1 and NCED3, were higher in OX plants than in the WT. Therefore, it would be desirable to analyze the levels of ABA in the WT and OX plants. 

Author Response

[Response to Reviewer 1 Comments]

This manuscript is describing the characterization of the AtBBD1 and AtBBD2 genes from the evidence that overexpression of AtBBD1 confers drought tolerance by enhancing the expression of regulatory genes in plant hormone abscisic acid (ABA)-mediated drought stress signaling in Arabidopsis. This manuscript also describes the identification of key regulatory genes in ABA-dependent pathway by using RNA-seq analysis in Arabidopsis plants overexpressing AtBBD1. This manuscript might be accepted for publication in International Journal of Molecular Sciences after the following points would be addressed.

  1. In Figure 2, the authors observed that the expression of AtBBD1 was at its peak at 3 h post treatment in the case of mock control. Therefore, it would be desirable to normalize the relative expression levels of both AtBBD1 and AtBBD2 genes by the levels of mock as a control.

[Response 1] We very appreciate the reviewer’s the insightful assessment and comments. In Figure 2, all samples at the point of 0 h were identical, that is, they were the untreated wild type. We already normalized the relative expression levels of both AtBBD1 and AtBBD2 genes by the levels at 0 h. Moreover, we used the same 8-fold scale of relative expression at Y-axis for all treatments.

  1. In Figure 7, two genes related to ABA biosynthesis, ABA1 and NCED3, were higher in OX plants than in the WT. Therefore, it would be desirable to analyze the levels of ABA in the WT and OX plants.

[Response 2] We agree with the reviewer’s opinion. In this manuscript, we characterized the physiological function of AtBBD1, a bifunctional nuclease, on drought stress and demonstrated that AtBBD1 functions as a novel positive regulator of drought responses by enhancing the expression of ABA- and drought stress-responsive genes as well as by increasing proline content. To our best knowledge, no nuclease has been reported yet to be involved in ABA-mediated drought resistance. However, further investigations, including changes of endogenous ABA level, are needed to elucidate the detailed mechanism(s) of AtBBD1 in ABA-dependent drought tolerance in our next manuscript, as the reviewer suggested.

Reviewer 2 Report

The work presented by Huque et al. is interesting and appears to be well performed. In this report, the authors explore the role of nucleases AtBBD1 and AtBBD2 in drought tolerance in the model plant Arabidopsis thaliana. They explore the connection of BBD1 and BBD2 with the ABA and JA, they analyzed different kind of reporter lines, checked proline content and assessed the expression of drought-related genes. They found that AtBBD1 functions as a novel positive regulator of drought responses by enhancing the expression of ABA- and drought stress responsive genes as well as by increasing proline content.

The work is interesting and worth publishing. The authors have to improve the statistical analysis. The text in general is well written, but it can be improved.

They authors use in the text Arabidopsis (written in italics) while it should be in italic written only full names as Arabidopsis thaliana or A. thaliana. The 'Arabidopsis' is a common name and shouldn't be written in italics. 

The authors should perform the statistical analyses on gene expression results as it is very important to show the statistical significant difference. 

The authors need to consider that they are usually analyzing a multivariate statistical problem (time, genetic background and expression levels/fold change in most occasions). Just using T-Test is incorrect. Appropriate statistical analyses need to be performed.

The RNA-seq analyses have been performed in replications also, for example Fig 8 B. All the data should represent the mean ± standard error values of three independent experiments with results of performed statistical tests. 

Author Response

[Response to Reviewer 2 Comments]

The work presented by Huque et al. is interesting and appears to be well performed. In this report, the authors explore the role of nucleases AtBBD1 and AtBBD2 in drought tolerance in the model plant Arabidopsis thaliana. They explore the connection of BBD1 and BBD2 with the ABA and JA, they analyzed different kind of reporter lines, checked proline content and assessed the expression of drought-related genes. They found that AtBBD1 functions as a novel positive regulator of drought responses by enhancing the expression of ABA- and drought stress responsive genes as well as by increasing proline content.

The work is interesting and worth publishing. The authors have to improve the statistical analysis. The text in genera is well written, but it can be improved.

Point 1: The authors use in the text Arabidopsis (written in italics) while it should be in italic written only full names as Arabidopsis thaliana or A. thaliana. The 'Arabidopsis' is a common name and shouldn't be written in italics.

[Response 1] We appreciate the reviewer’s valuable suggestions and comments. As the reviewer suggested, we have changed all “Arabidopsis” to gothic (normal).

Point 2: The authors should perform the statistical analyses on gene expression results as it is very important to show the statistical significant difference.

[Response 2] We agree with the reviewer’s opinion. When the difference of gene expression revealed more than two-fold, we did not perform the statistical analyses. In this revision, however, as the reviewer suggested, we have analyzed the one-way ANOVA and replaced Figures 4, 7 and 8 with new figures showing the statistical significance.

Point 3: The authors need to consider that they are usually analyzing a multivariate statistical problem (time, genetic background and expression levels/fold change in most occasions). Just using T-Test is incorrect. Appropriate statistical analyses need to be performed.

[Response 3] Thank you for pointing out this. As the reviewer commented, we have analyzed the one-way ANOVA for qRT-PCR data, because multivariate statistical analyses cannot be applicable in our gene expression experiments.

Point 4: RNA-seq analyses have been performed in replications also, for example Fig 8 B. All the data should represent the mean ± standard error values of three independent experiments with results of performed statistical tests.

[Response 4] Thank you for pointing out this. Our initial purpose for RNA-seq analysis was to screen some genes differentially expressed by the alternation of AtBBD1 expression, not to analyze the whole transcriptomic changes. Accordingly, we analyzed RNA-seq in a single run and validated our RNA-seq expression data by qRT-PCR with three independent biological replications. Thus, we performed the statistical analyses for qRT-PCR, but not for RNA-seq data. For example, the RPKM values of RNA-seq (Figure 8B) were validated by qRT-PCR (Figure 8C).
